# Aluminum Alloy Reinforced with Agro-Waste, and Eggshell as Viable Material for Wind Turbine Blade to Annex Potential Wind Energy: A Review

**Imhade P. Okokpujie** [1,2,*] and **Lagouge K. Tartibu** [1]

1 Department of Mechanical and Industrial Engineering Technology, University of Johannesburg, Johannesburg 2028, South Africa; ltartibu@uj.ac.za
2 Department of Mechanical and Mechatronics Engineering, Afe Babalola University, Ado Ekiti 360101, Ekiti State, Nigeria
* Correspondence: ip.okokpujie@abuad.edu.ng

**Abstract:** The most important part of the wind turbine is the blade. From existing studies, it has been concluded that most wind turbine blades have a high rate of failure during operation due to fatigue, because of a lack of proper material selection processes. Materials such as fiberglass, wood, aluminum, and steel have been used but have not been able to qualify as sustainable materials. Therefore, this study focuses on the review of existing materials employed for developing metal matrix composites as ecological materials to produce wind blades. This study discusses the application of aluminium, silicon, and magnesium metal matrix alloys and the implementation of agro-waste materials (coconut rice, coconut shell, rice husk ash, and sugar Bagasse ash) and eggshell as reinforcement particulates for metal matrix composites for developing wind blades. The study also reviews the method of production of matrix composites. From the results obtained via the review, it is clear that the application of eggshells assists as a binding element for proper mixture, and the combination of Al–Si–Mg alloy with coconut rice and shell improves the strength of the material, since wind blades need durable materials and ductility due to their aerodynamic shape to convert enough energy from the wind.

**Keywords:** aluminum metal matrix composite; silicon carbide; magnesium; agro-waste; eggshell; wind blade

## 1. Introduction

Aluminum metal matrix composite (AMMC) is a class of materials with aluminum as the base metal that has successfully met the most rigorous specifications in applications where lightweight, high stiffness, and moderate strength are the requisite properties. A wind turbine is a device that converts the wind's kinetic energy into electrical energy [1]. The wind turbine blade is the part of the turbine that captures the wind and transfers its power to the rotor hub for electricity generation. Wind energy is one of the greenest forms of energy. Coupled with the fact that it is renewable, it is constantly alluring to investors internationally. Indeed, these investors would not want to take uncalculated risks by investing in something they are not certain will succeed [2], ergo, they employ the services of engineers to seek out the best ways of ensuring their investment in wind energy generation is secure. One of the numerous ways engineers do this is by ensuring the wind turbine blade, described in the previous chapter as an essential part of the system, is made of quality material [3]. Over time, composites have proven to be better than non-composites due to the many advantages they pose over them, including higher corrosion resistance, higher durability, excellent heat resistance, lighter weight, better performance, and high stiffness. One major setback and baffling aspect of wind energy applications are finding a suitable or somewhat perfect material to construct the blade. This has led to setbacks

for manufacturers in this field and even other members of society [4]. Several research studies, however, have been conducted to demonstrate and conclude the critical need for improvement in the material selection of wind turbine blades. Reinforcements are supplements added to a monolithic element such as aluminum or even composites to help improve their properties. Mohanavel et al. [5] use agricultural by-products, elements, and a compound as reinforcements. Since wind power is a vast engineering field, copious studies have been conducted to facilitate disruption in the industry since any prosperous circular economy can only exist when it relies solely on renewable energy sources [6]. The ever-rising global desire to eliminate fossil fuel usage has led scientists to seek better ways for energy generation. It is impossible to harness the wind's incredible energy without employing a wind turbine. Additionally, wind turbines cannot operate without wind turbine blades since they are the ones that trap the wind in the first place. Consequently, the wind turbine blade must be efficient and of high durability to help bolster its performance and extend its life.

In searching for the best characteristics that the wind turbine blade must possess to achieve this feat, scientists have conducted several types of research by comparing the characteristics of materials and elements to their desired features. Finally, they discovered that no single element or material possesses all the characteristics [7]. This then introduced composites, which had been investigated earlier. Section two of this study is a review of some research done on different materials, elements, compounds, and matrix composites and why they have been considered to have properties that qualify them for use in manufacturing wind turbine blades [8]. The materials used in a sustainable wind blade must be able to withstand various forces during its application due to the aerodynamic shape required for adequate energy conservation. This shape and the forces acting on the blade are presented in Figure 1. However, the focus of this research is on the long-term viability of Al–Si–Mg alloy reinforced with agro-based and eggshell as standard materials for the development of wind turbine blades. This review examined aluminum alloy, silicon, magnesium, eggshell, coconut shell, coconut rice, bamboo leave ash, and others as reinforcement materials.

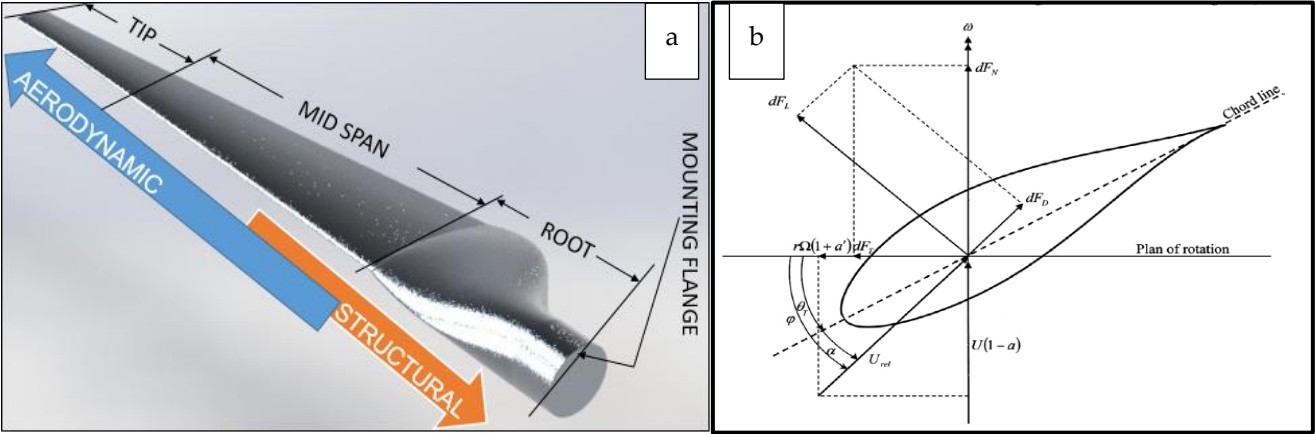

**Figure 1.** (**a**) The aerodynamic shape of the blade and (**b**) the analysis of the forces acting on the wind turbine blade via the wind speed.

## 2. Metal Matrix Composite Materials as a Sustainable Material for Wind Turbine Blade Production

Now, considering wind as a source of energy, the wind turbine, as said earlier, is a significant component. It is critical for the production or manufacture of a wind turbine to use materials that can last a long time, resulting in a higher profit with lower maintenance costs [9]. Metal matrix composites have been proven to offer several advantages over ordinary materials. In other words, all the advantages they offer include flexibility in design, improved productivity, greater tolerance, higher corrosion resistance, and higher

durability, which means, excellent heat resistance, lighter weight, better performance, and high stiffness [10]. Moreover, better performance under extreme conditions such as elevated temperatures and better chemical resistance ensures the blade lasts longer. An important consideration in the production of a horizontal axis wind turbine blade for green energy generation is the material choice procedure when considering developing countries such as Nigeria because of the challenges caused by the low wind speed variations [11]. Because wind turbine edges are so complex, the materials used for sharp edge advancement must be machinable, strong, high erosion- and wear-rate safe, of high solidity to lightweight proportion, and financially prudent. This is the rationale behind exploring studies utilizing different techniques to examine material decisions for planning wind turbine cutting edges [12]. There are different strategies to carrying out the material determination process; be that as it may, this review centres around the multi-criteria design method (MCDM). MCDM is a fantastic tool for dealing with complex issues in design and other fields of life that deal with the issues of [13]. The method aids in segmenting the problem, and the chief investigators use it to identify the problems and suggest workable remedies. AHP, data envelopment analysis approach, entropy technique, TOPSIS, EDAS, FAHP TOPSIS, TOPSIS leveraging Excel, and WASPA are some of the numerous strategies that are included in multi-criteria decision making. It is stable and incredibly simple to use [14,15].

The results from the four other choices—alloy, metal, glass fiber, and mild steel—in a study by Okokpujie et al. [16] during the component estimation interaction of a wind turbine blade advancement using the MCDM (AHP and TOPSIS) showed that the aluminum integration has the best display worth of 78%, followed by 67% for glass fiber, 44% for treated steel, and 25% for mild steel. After conducting a thorough analysis, the authors recommended that manufacturers of wind turbine edges use the aluminum 6061-T9 alloy for improvement since it has excellent erosion resistance, is durable, has a high strength-to-weight ratio, and cannot react to air. From this, it is seen that aluminum alloy (a matrix having aluminum as its main constituent) is proven to be better even than aluminum alone.

From the above Figure 2, the aluminum alloy, alias aluminum metal matrix, is comparatively better than stainless steel, glass fiber, and mild steel performance-wise, with a rating of 0.79. As shown, it is far better than mild steel (a metal considered very good for engineering applications). This is evidence that AMMCs are sustainable materials for engineering applications, even in wind turbine blade manufacturing (which is an engineering application). Another analysis for comparison is comparing some of the important qualities they possess including durability, corrosion resistance, and densities. Once again, AMMCs dominate the other materials in this category; thus, they are arguably better than mild steel, glass fiber, and even stainless steel [17]. This property is shown in Figure 3.

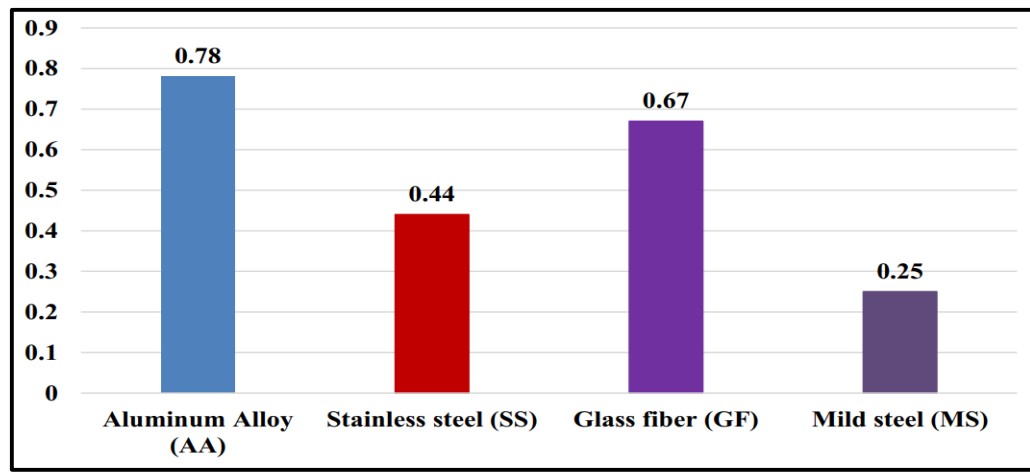

**Figure 2.** The performance value analysis of the alternative materials for the wind turbine blade.

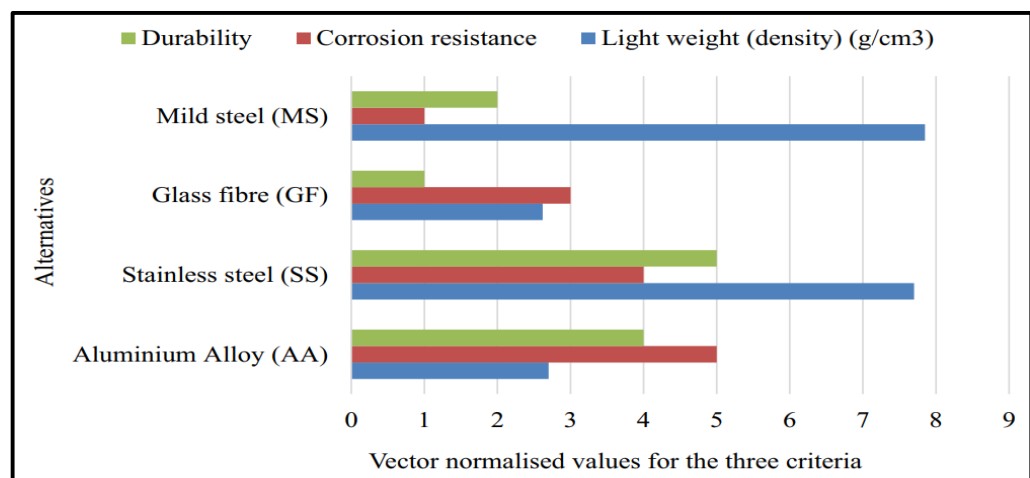

**Figure 3.** The analysis of the alternatives with the normalized vector values.

Metal matrix composites are made up of two or more different materials, including at least one metal and another material such as ceramics or organic compounds. They are specifically referred to as "hybrid composites" [18]. To realize the optimum blend of properties in composites, they ought to be fabricated by taking charge of the morphologies of their constituents, since the properties of any composites rely upon their synthetic arrangement or the properties of their constituent stage math, including their particles' size, shape, and direction in the framework [19]. Metal grid composites (MMCs) are advanced materials with numerous applications, including gadgets, aviation, thermal energy stations, vehicle enterprises, security, brandishing ventures, bio-medical devices, and so on.

Aluminum composites are primarily supported by utilizing hard materials such as boron carbide ($B_4C$), silicon carbide (SiC), boron nitride ($B_4N$), alumina ($Al_2O_3$), AlN, $TiB_2$ $TiB_2$ as well as natural fortifications such as fly debris. Scraped spots, high sturdiness, worked on warm conductivity, low thickness, higher weakness perseverance, solidness, higher solidarity to weight proportions, machinability, opposition, creep obstruction, dimensional steadiness, and better high-temperature execution [20] are some of the enhancements that can provide significant advantages over base metal composites. A few creation processes are utilized for the planning of aluminum metal framework composites (AMMCs). The extensively utilized standard cycles incorporate mix projecting, powder metallurgy, and crush projecting [21]. The mix projecting strategy is, in general, utilized for support on account of its straightforward entry when contrasted with other techniques. Mix projecting cycles improve the holding strength between the supported particles and the grid due to their increased mixing activity [22]. The key test confronted with mix projecting is the isolation or cleaning of built-up particles on the grounds that, after wetting, a few particles sink and some buoy because of thickness contrasts during the hardening system and, accordingly, different projecting deformities such as blow openings and porosities.

The support utilized in the metal framework influences the strength and solidity of the material in contrast with its constituent parent materials. The current situation straightforwardly interfaces the mechanical field and its examination with the auto and aeroplane industries [23]. Aluminum's limitation is that it is easily scratched and dented because it is a very light and similarly weak material. Hence, there is motivation to create an aluminum-based composite that will supplement aluminum's deficiencies by being wear-safe through the expansion of appropriate fortifications to a characterized extent. Aluminum-matrix composites have a phenomenal capacity to bear elastic and compressive powers [24].

In a recent study on the impact of fly debris particles with aluminum softening on the wear of aluminum metal network composites [25], the following was observed:

- Fly ash contents are effectively used in the production of the AMMC;

- The dispersion of the fly debris parts all through the aluminum lattice was even;
- The wear resistance of the manufactured composites increased with the increase in fly ash contents.

The example with medium fly debris content (4%) brought about the most minimal normal coefficient of grinding (0.12), and the example with high fly debris substance (6%) shows the greatest normal coefficient of grinding (0.161) [26]. This was proof that the more fly ash was added, the greater the rise in the friction coefficient between tribo-pairs [27,28]. Some observations were made from the analysis. Figure 4 shows a relationship between the wear properties of the matrix and the amount of reinforcement used.

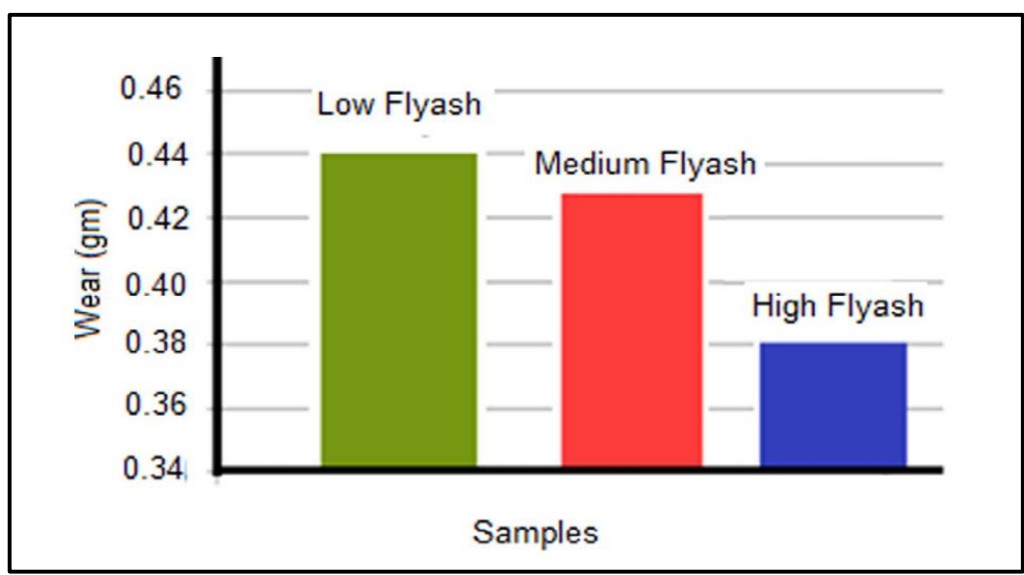

**Figure 4.** Variation of wear for low, medium, and high-content fly ash specimens [27].

All the above observations indicate that metal matrix composites are better than ordinary metal in engineering applications [27]. Another research study was recently conducted on aluminum 7075 (Al-7075) metal matrix composites to check their mechanical properties, tribological properties, and consumption conduct by adding positive fortifications. Beginning around 1943, aluminum 7075 series amalgam materials have been utilized for the development of helpful parts of the auto, aviation, and marine applications, inline skating outlines, shafts for lacrosse sticks, hang lightweight flyers, airframes, rifles for the military, utensils, fuselage, turbine packaging and parts, rocket tail cone, bike parts, car motor packaging and a large group of others. These materials have a high strength/weight proportion that are just as lightweight and strong. In different fields, all-inclusive necessities are elite execution rate, minimal expense, and better-quality materials made by analysts that exchange from solid to composite materials [29].

Faisal et al. [30] investigated how fluid metallurgy strategies with different particulate weight divisions (2.5, 5, and 7.5%) were used to create aluminum–boron carbide composites. In this examination, a stage distinguishing the proof was performed on boron carbide utilizing x-beam considerations. A microstructure investigation was likewise finished with SEM, and composites were described by hardness and pressure tests. The results revealed an easy increase in the size of the boron carbide and a decrease in thickness while the hardness increased. Other recognizable changes include an increment in the compressive strength of the composites and an increment in the weight level of the boron carbide in the composites. When compared to having a MMC weighed down with one more component or amalgam, in this case boron carbide, this is a fantastic strength-to-weight ratio. All continuously supported composite materials show anisotropic qualities. Their properties depend upon the direction, which varies with various crystallographic directions [31].

It is not a new discovery that the mechanical properties of aluminum alloys as composites improve with increasing reinforcement materials. This implies that with the addition of reinforcements to an AMMC, for instance, there is an increase in its strength and stiffness since its strengths are combined [32].

### 2.1. Aluminum Alloys as Sustainable Reinforcement Material and Based Materials

Aluminum is a white, shining metal and one of the most abundant elements on Earth. It takes up approximately 8% of the Earth's surface mass. One cannot find pure aluminum, as it is very reactive with other elements [33]. The most common form of aluminum is aluminum sulfate. Some earth minerals react with $H_2SO_4$: lithium, potassium, and calcium aluminum [34]. Aluminum sulfates are employed in cooking, cosmetics, and the like. It appears that aluminum got its name from aluminum sulfates (alumen in Latin). The physical properties of aluminum that are commonly used are its electrical and thermal conductivity, its tender texture, and its treatable surface. Aluminum is generally applied in the various fields of automotive, marine, and aviation, as well as other objects that require lightweight materials. The physical and mechanical properties of pure aluminum include phase (at stp): solid; melting point of 660.32 °C; density of 2700 kg; yield strength of 3.5 MPa; Young's modulus of 70 GPa; Mohr's modulus of 275; boiling point of 2470 °C; shear modulus of 26 GPa; Vickers hardness of 160–350 MPa; Brinell hardness of 160–550 MPa; ultimate tensile strength of 11 MPa; Bulk's modulus of 76 GPa; and elongation of 40% in 50 mm [35].

Aluminum goes through a series of processes before it is changed from its archetypal state to a purer state. This shows that the production of aluminum is extreme and necessitates massive amounts of energy. Aluminum can be reused by dissolving it in the same way it may be used to rebuild matching items [36]. The rocky state of aluminum ore is known as Bauxite. Cryolite ($Na_3AlF_6$, sodium hexafluoro aluminate) is another source of aluminium. It used to be extracted from cryolite (before 1888, when an Austrian chemist, Carl Josef Bayer, invented the Bayer process of extracting alumina from bauxite) [37]. Bauxite contains 30-60% aluminum oxide ($Al_2O_3$), silica, iron oxides, and $TiO_2$. The bauxite ore is given in Figure 5.

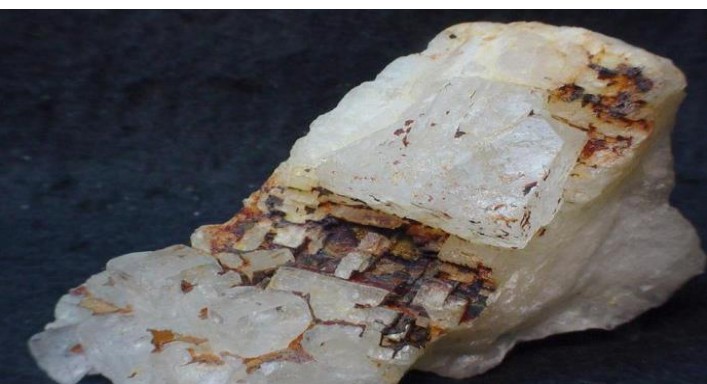

**Figure 5.** Bauxite ore.

Aluminum is the most widely available metal in the earth's crust, accounting for about 8% of the earth's solids [38]. However, aluminum exists in a combined form, usually as the hydrated oxide of bauxite [39]. Aluminum is very active, but a protective aluminum coating or film is developed that protects it from corrosion in many environments [40]. The advantageous characteristics of aluminum are its lightweight, corrosion resistance, ease of fabrication, and appearance [41]. Aluminum is alloyed with small quantities of elements, which convert the soft, weak metal into a complex and robust metal while still retaining its light weight, which makes it useful for engineering purposes and industrial applications. The major alloying elements used with aluminum are copper, magnesium, zinc, manganese, silicon, nickel, iron, and titanium. Other elements are added for unique

properties. Aluminum that has been properly alloyed and treated may resist corrosion brought on by the sea, salt, and other climatic influences, as well as a broad range of other physical and chemical contaminants [42]. The ease of fabrication of aluminum alloy into any form is one of its most critical assets. It has a low degree of workability and can be cast by any method. It can be rolled into any thickness, like thin foil. Aluminum products subjected to plastic deformation by hot and cold working processes such as rolling, extrusion, and drawing to obtain the desired products result in microstructural changes caused by the five working processes, which are accompanied by thermal treatments used to control some properties and characteristics of the wrought aluminum alloy [43].

Aluminum is extracted from its bauxite ore via either the Bayer process or the Hall–Heroult process [44]. The Bayer process is a standard technique used by industries to extract aluminum and alumina from bauxite, as shown in Figure 6. The bauxite has 10–30 wt.% $Fe_2O_3$, 4–8 wt.% $SiO_2$, and 2–5 wt.% $TiO_2$ as its major constituents, which are further pulverized and dissolved in NaOH. The filtration process separates the impurities, while alumina hydroxide is precipitated to get the alumina, as shown in Equations (1)–(3).

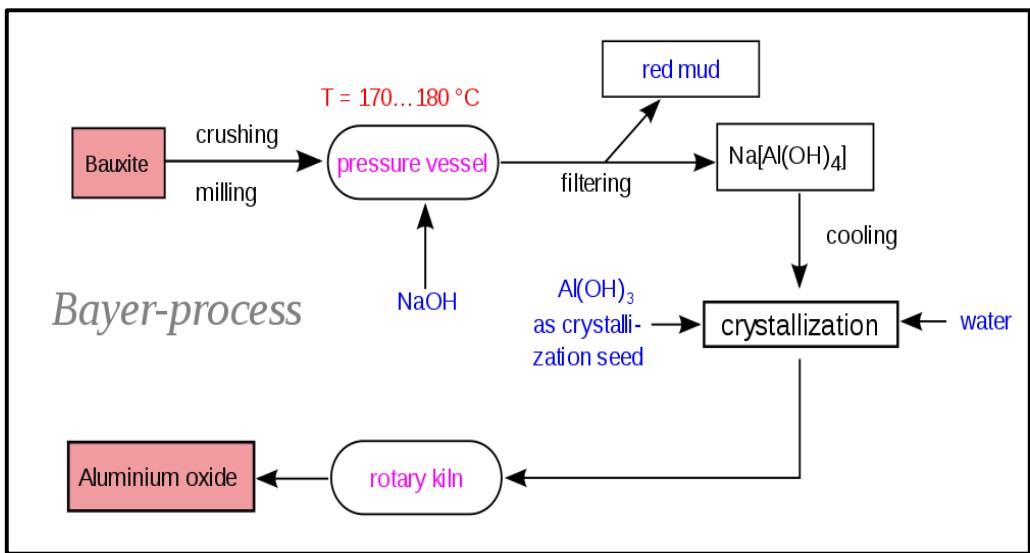

**Figure 6.** The Bayer process [44].

$$Al_2O_3 + 3H_2O + 2NaOH + HEAT \xrightarrow{} 2NaAl(OH)_4 \quad (1)$$

$$2Al(OH)_3 + HEAT \xrightarrow{} Al_2O_3 \quad (2)$$

$$2Al_2O_3 + 3C \xrightarrow{} 4Al + 3CO_2 \quad (3)$$

The demand for magnesium-based MMCs has steadily increased because of the growing demands for lightweight and high-performance materials. The MMCs based on magnesium alloys, particularly Mg–Al systems, are a better option for manufacturing or engineering light structural materials [45]. Their use in disc rotors, piston ring slots, gears, transmission bearings, crankshafts, wind turbine blades, armaments, and other components has a significant potential for use in construction, automotive, aerospace, and military applications. Additionally, increasingly complex and technological processes like the stir casting method necessitate knowledge, making it challenging to use "incompetent" labour from nations such as Nigeria. This class of reduced-density materials can be introduced to the market by using inexpensive materials, alloys, and reinforcements [46]. This shows that heat treatment is certifiably not a compelling way of working on the mechanical properties of these amalgams. Thinking about this, consolidating fortifications that are thermally stable into composite materials permits them to be appropriate for high-temperature

applications. Mg–Al amalgams, for example, AM60 and AZ91, are by far the most common magnesium composites used in the auto industry [47]. They also serve as the most widely used framework for materials reinforced with magnesium. Other magnesium compounds have also been used as lattice materials, but less frequently, including pure magnesium, Mg–Li composites, and Mg–Ag–Re (QE22) compounds. The most widely used support for magnesium-network composites is ceramic granules. Clay components are desirable for fortifications due to a few of their common characteristics, which incorporate low thickness and high hardness levels, strength, flexibility, and warm security [48]. Notwithstanding, they likewise have some normal limits, such as low wettability, low pliability, and low similarity with a magnesium grid. Among the different ceramic fortifications, silicon carbide (SiC) is the most well-known in light of its moderately high wettability and its magnesium liquefaction security contrasted with different pottery [49].

Yadav et al. [50] concentrated on the machinability and mechanical properties of Al MMC supported by SiC particles. With the expanded weight rate, the SiC hardness, rigidity, and thickness of Al MMCs expanded while durability diminished. The authors additionally concentrated on the effect of aluminum lattice on conductivity and built up SiC particles exposed to various temperature conditions. The outcomes showed that by isolating the particles, the conductivity of composites was influenced. Shukla et al. [51] investigated the high cycle fatigue and studied the fracture behavior of 7034/SiC/15P-4A and 7034/SiC/15P-4A metal matrix composites (MMCs). Increasing the temperature led to a proportional decrease in the strength modulus, microstructure, and ductility of the composites. There was a greater potential for fatigue due to an increase in the load ratio.

Particle segregation's impact on Al MMCs enhanced by SiC particle flow patterns was studied by Shen et al. [52]. The findings demonstrated that particle clustering had a greater impact on the matrix's mechanical properties than on its elastic properties. Particle segregation occurs during the tensile deformation test, and micro-structured particles encounter a larger percentage of particulate rupture than uniformly dispersed nanoparticles. The variables affecting the surface finish on the machining of Al/SiC granular MMCs were investigated by Palanikumar and Karthikeyan [53]. To achieve the small surface roughness, normal probability plot, response table, and graph, they made sure all the process variables, such as spindle speed, were as efficient as practicable. Chuan-bo et al. [54] investigated the impact of a 3.5% NaCl thermal treatment on the corrosion protection of the 6061 aluminum alloy. Although a drop in hardness enhanced the alloys' resistance to corrosion, it was found that the solution-treated 6061 aluminum alloys had increased hardness but improved corrosion resistance.

However, most aluminum alloys are mostly made up of aluminum, silicon, and magnesium elements, and others. Therefore, there is a continued need to discuss the reinforcement capability of silicon and magnesium as sustainable reinforcement materials.

### 2.1.1. Magnesium as a Sustainable Reinforcement Material as a Major Element in Aluminum Alloy

Magnesium has excellent corrosion resistance when exposed to a marine atmosphere, making it useful in food and beverage processing. Its moderate to high strength and toughness make it frequently used in architectural and other decorative applications. It has good weldability, is machinable, polished, or anodized [55]. One disadvantage of magnesium MMC is that it is relatively expensive due to the more complicated manufacturing processes required. Since it has high corrosion resistance, the acids in the atmosphere, oxygen, and other corrosion agents will not damage materials. This means that if magnesium is used to reinforce an aluminum alloy, it will improve its corrosion resistance [56]. The product of alloying magnesium with aluminum is used to manufacture a wind turbine blade, which will cause the blade to resist corrosion as it rotates to trap the wind. Further, since magnesium has beneficial attributes in marine environments, it can be used to manufacture blades that would be used for offshore wind turbines. developing a vast range of processing techniques and reinforcements, of which high-performance magnesium materials are a part.

The addition of magnesium to aluminum–silicon will give the alloy casting characteristics that enable it to respond well to heat treatment and help retain a reduced level of thermal expansion. The magnesium-based MMCs, which are unit-directional and have a density as low as 1.8 g/cm3, can easily demonstrate a bending strength of 1000 MPa [57,58]. It is possible to maintain the improved mechanical quality at temperatures as high as 350–400 °C [59]. The creation of a composite might be the only method for reinforcing some magnesium alloys. HCP (a) and BCC (b) solid solution phases, for instance, make up Mg–Li binary alloys with eutectic composition. Without producing any Mg–Li precipitates during cooling, Li's dissolution into Mg has a partial solid solution-strengthening effect [60].

2.1.2. Silicon as a Sustainable Reinforcement Material as a Major Element in Aluminum Alloy

As established earlier, aluminum composites are mainly reinforced using hard materials such as carbide ($B_4C$), silicon carbide (SiC), boron nitride ($B_4N$), alumina ($Al_2O_3$), AlN, $TiB_2$, as well as organic reinforcements such as fly ash [61]. From this statement, it is evident from the literature that SiC is a top-notch choice for reinforcing aluminum alloys. One of the most advanced ceramics in use today is silicon carbide (SiC). SiC ceramics are widely used as lightweight armour ceramics, having a hardness of 25 GPa and a low density of 3.21 g/cm$^3$. Ceramics are also used to produce specialized automotive, aerospace, and nuclear reactors [62]. It exists naturally as moissanite. Conjoined silicon carbide creates tough ceramics, which are generally employed for use in high-durability applications such as vehicle brakes, clutches, bulletproof-vest ceramic panels, and even wind turbine blades. SiC particles show a higher hardness than aluminum [63].

Silicon carbide (SiC) offers excellent wear resistance and mechanical qualities, such as high-temperature strength and thermal shock resistance. Silicon carbide is produced when silicon interacts with carbon, forming SiC [64]. Sintered SiC is made using non-oxide sintering aids and pure SiC powder. The material is sintered in an inert atmosphere at temperatures of up to 2000 °C or higher, using conventional ceramic forming techniques. Silicon carbide (SiC) retains its excellent mechanical strength even when exposed to extreme temperatures as high as 1400 °C. It is also more corrosion-resistant than other ceramics. According to Yanzhen et al. [65], here are the physical and mechanical properties of the SiC ceramic: maximum service temperature, 13,800 °C; density, 3.02 g/cm$^3$; open porosity, 0.1%; bending strength, 250 MPa at 20 °C and 280 MPa at 1200 °C; elastic modulus, 330 GPa at 20 °C and 300 GPa at 1200 °C; thermal conductivity, 45 W/mK at 1200 °C; thermal expansion coefficient, 4.5K-1 $\times$ 10$^{-6}$; Mohs' hardness, 13; acid and alkali resistance, excellent [66] reviewed the effect of reinforcing aluminum with silica carbide, where the scanned electron microscope of various composition percentages of aluminum–silicon carbide matrix composite was conducted on Al 85% and SiC 15% and possessed the highest strength. The implication of this is that it lasts longer than many other industrially applied matrix composites and has a high amount of stress.

All the investigated composites' abrasion wear durability is greatly increased by the addition of alumina fortification, and the fortified aluminum alloys exhibit significantly greater wear resistance [67]. $Al_2O_3$ content in the study boosted wear resistance as well. Because $Al_2O_3$ particle content and structure within the range investigated increased, so did the wear resistance of the composites. Even so, $Al_2O_3$ particle size had a greater impact on the composites' wear rate than did $Al_2O_3$ particle concentration. The active resistance of the $Al_2O_3$ particles to penetrating, cutting, and polishing by the SiC abrasive sheets was a major factor in the composites' remarkable friction coefficient [68].

Bakhtiari et al. [69] demonstrated in their study of ferrosilicon addition and the in-situ formation of SiC nano-whiskers on MgO–C refractories that the addition of ferrosilicon significantly increased the strength of the alloy. Mogale and Matizamhuka [70] observed that upon adding SiC to $Al_2O_3$, the SiC grains were ingrained into the $Al_2O_3$. The study shows that $Al_2O_3$–SiC composite is an innovative material that could improve wear resistance with a volume percent above 15. Even with this excellent discovery, the reinforcement with

SiC did not improve the corrosion resistance. It is worth noting that this element can be obtained from agro-waste materials. Therefore, some selected agro-waste materials will be investigated to annex their potential as reinforcement materials.

### 2.2. Coconut Rice and Coconut Shells as Sustainable Reinforcement Materials

The coconut is a giant fruit that grows on a palm tree and has a hard shell containing liquid and white meat. They are and have been helpful for countless years. Their shells have great strength and elasticity. They are also of high density and desirable specific gravity [71]. Their rough surface texture provides a grip area for other materials if used as reinforcement in a matrix. Coconut (Cocos nucifera) is a member of the palm family Arecoceae. Its genus is Cocos. It consists of the coconut palm, coconut fruit (drupe), coconut seed, and coconut shell containing water. The cocoanut, as it was called in ancient times, originates from coco, a Spanish and Portuguese word referring to a head or skull because the three holes on its shell look like a face. Prasanna et al. [72] conducted an experiment on soil reinforcement using coconut shell ash in a case study of Indian soil. From the results obtained, it was seen that the expansion of 0.4% to 0.8% of coconut shell debris exhibited the most extreme improvement in dry thickness and ideal dampness content, and, furthermore, the point of inside grating and attachment. Ergo, coconut shell debris could be utilized as a very outstanding waste material for soil support under the review region, which falls under the area of South Goa in the Taluka of Salcete in India. By contrast, from every one of the after-effects of Atterberg's cut-off points for Test 1, it very well may be presumed that the most extreme versatility record, fluid breaking point (approx.), was accomplished at 2%, and the greatest plastic breaking point was obtained at 10% coconut shell debris support. In Test 2, the most extreme pliancy record, fluid cut-off, and plastic breaking point were obtained with 5% coconut shell debris as support.

Agunsoye et al. [73] experimented on the mechanical conductivity of a coconut shell-based polymer framework composite. The outcome showed that the hardness of the composites expanded with an expansion in coconut shell content, but the rigidity, modulus of flexibility, sway energy, and malleability of the composite diminished with an increment in the molecule content. The SEM of the composites' (with 0–25% particles) surfaces shows a not so good interfacial association between the coconut shell molecule and the low-thickness polyethylene lattice. This review, therefore, takes advantage of the capability of agro-based waste fibre in Nigeria as an elective particulate material for fostering another composite. From the experiment, the following conclusions were drawn:

- The non-homogeneous dispersion in the microstructure of the coconut shell-supported polyethylene composite is the main consideration liable for the lessening in strength when contrasted with the control test having no coconut shell particles;
- As the level of the coconut shell particles expanded, there was a corresponding decline in porosity along these lines, making the composite appropriate for the application in the inside piece of an engine vehicle where materials with great hydrophobic qualities are required;
- Coconut shell particles further developed the hardness property of the polyethylene framework composite. This characteristic is an additional prerequisite for vehicle interiors;
- Since matrix composites are considered usable in the automobile industry, they can also be employed for manufacturing wind turbine blades since they are also lightweight.

Madakson et al. [74], in a work on characterizing coconut shell ash for potential utilization in MMCs for automotive applications, said that the coconut shell ash possesses about the same chemical phases as other reinforcements including rice husk ash and fly ash, which have been used in metal matrix composites (MMCs) primarily for automobile applications. It was concluded that coconut shell ash could be used as a low-cost reinforcement in metal matrix composites (MMCs). In this research, the mechanical characteristics of hybrid matrix composites made of aluminum 7075 alloy, boron carbide, coconut shell, and fly ash were studied [75]. The aluminum 7075 alloy was reinforced with boron carbide ($B_4C$), coconut shell, and fly ash particles. It was concluded that the uniformly distributed

B$_4$C, coconut shell, and fly ash particles added as reinforcement in the Al7075 alloy helped improve the composites' impact strength, hardness, and tensile strength. Another study evaluated the mechanical properties of biodegradable coconut shells and rice husk powder as polymer composites for lightweight applications. It showed that agro-based reinforced composites are inexpensive, can be recycled, and are lightweight, eco-friendly materials. They have desirable flexural and tensile properties and can be used in place of conventional fibers, including glass [76].

Kumar et al. [77] looked up the mechanical properties and characterization of aluminum (Al 6082) reinforced with zirconium oxide and coconut shell ash. It was observed that upon adding ZrO$_2$ and coconut shell ash, the hardness of the hybrid composite increased. Further, the tensile and yield strengths of the hybrid composite appreciated in value by 23.17% when ZrO$_2$ was added and by 26.78% when coconut shell ash was added. This yet again proves the superiority of AMMCs over monolithic aluminum. Figure 7a,b show the coconut rice and the elemental composition from the energy dispersive X-ray spectroscopy, which proves it has a high level of carbon elements, which in developing MMC can assist in improving the hardness and strength of the materials. Figure 7c shows the coconut shell.

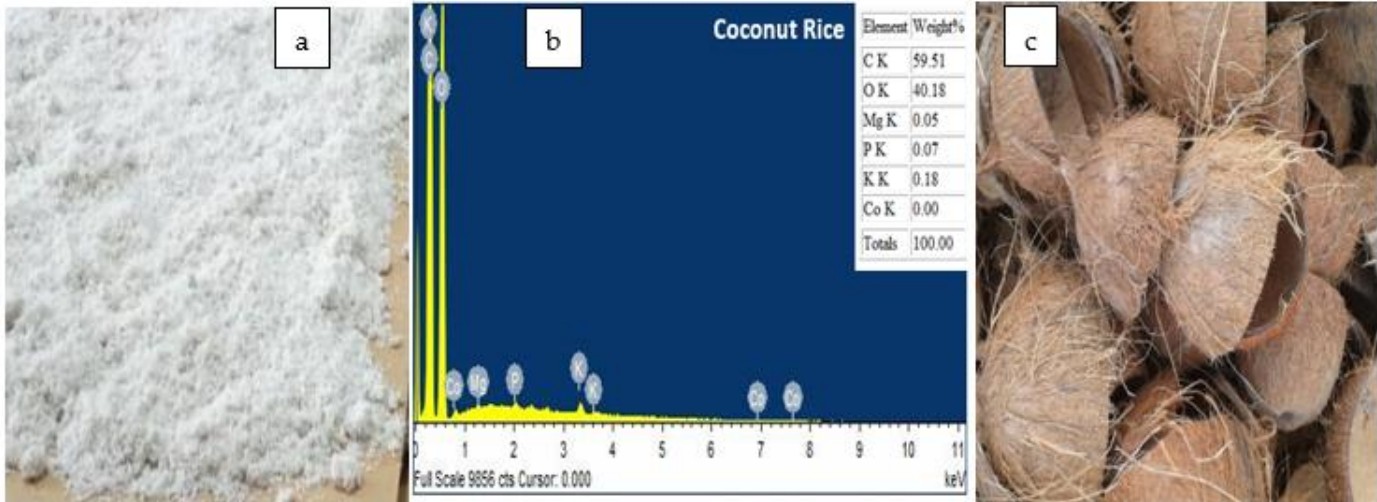

**Figure 7.** (**a**) Coconut rice, (**b**) elemental composition of coconut rice, and (**c**) coconut shells.

### 2.3. Eggshells as a Sustainable Reinforcement Material

Malik et al. [78] experimented with the fabrication and mechanical testing of eggshell particle-reinforced AlSi composites. It was found that the different mechanical properties of the subsequent composites, such as rigidity, modulus of versatility, hardness, sturdiness, sway, and compressive qualities, increased upon the addition of the eggshell particles, establishing that the consolidation of waste eggshell particles in the AlSi matrix serves as reinforcement. The increase in mechanical properties such as the tensile (6.61% UC, 10.61% C), the hardness (10.2% UC, 19% C), the impact strengths (30.07% UC, 302.35% C), and the compressive (9.12% UC, 63.94% C) was more prominent in the presence of carbonized eggshell particles. Field-emission scanning electron microscopy was also conducted to ascertain the constituents of the materials in the matrix and the composites.

According to Ononiwu et al. [79], aluminum matrix composites (AMCs) are becoming more common because of their improved mechanical and corrosion-resistant properties. This study was conducted to determine the effect of carbonized eggshells on the morphology, density, and resistance to corrosion of AA 1050. Because they are plentiful, simple to produce, and most significantly, environmentally sustainable, carbonized eggshells were selected as the reinforcement agent. The method of creating composites that was chosen was stir casting. Adjustments were made to the weight percentage of carbonized eggshells in the aluminum alloy, which ranged from 2% to 8%. The aluminum matrix's reinforcements were evenly

dispersed throughout, according to the structural investigation. The ability of the process to create lightweight parts was demonstrated by the study of the experimental density produced via the Archimedes' principle, which showed a decrease in density with rising eggshell weight fraction. Tensile strength research revealed improvements of 11.04% for the 2 weight percent and the 8 weight percent, respectively. The micro-hardness increased, and the least corrosion rate was attained when the weight percentage of the carbonized eggshell particles rose to 8 wt%. Figure 8a–c depict raw eggshell, processed eggshell, and the EDX to show the chemical composition of the eggshell, as well as an eggshell power sample and the elemental composition of the eggshell from energy dispersive X-ray spectroscopy. When used for AMMC, eggshell has a silicon element content of 63.7%, according to the EDS in Figure 8c. It helps to promote corrosion resistance, enhance wear characteristics, and aid in appropriate formulation throughout the preparation process.

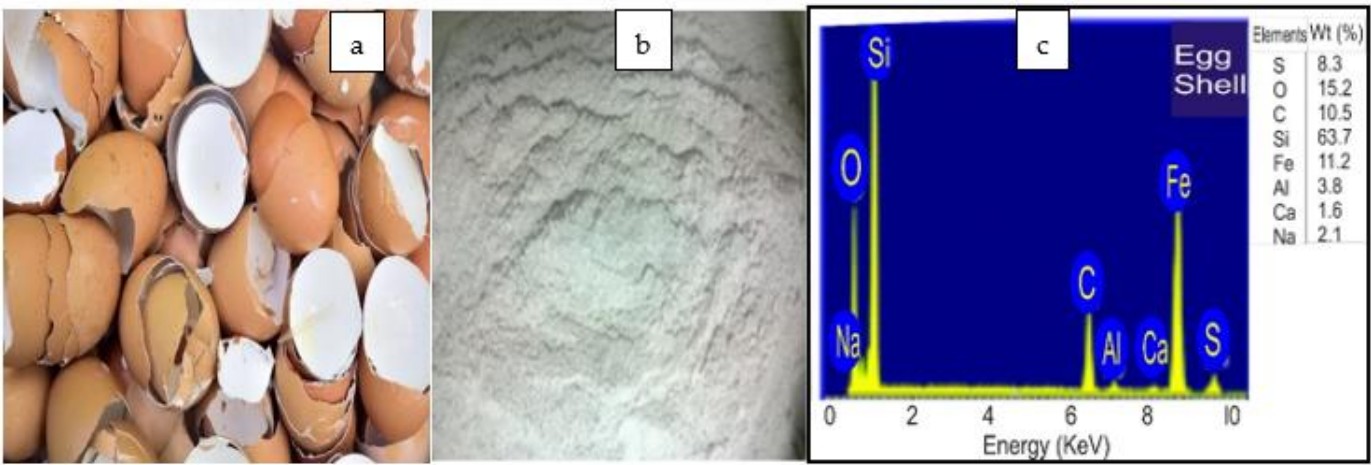

**Figure 8.** (**a**) Eggshell, (**b**) raw and powder, and (**c**) elemental composition of eggshell.

Hussein et al. [80] explain that, although magnesium and magnesium alloys are biodegradable and have low elasticity, the mechanical and corrosion properties of pure magnesium are impacted by its quick breakdown in the body. In the current work, powder metallurgy was used to create the Mg and magnesium 1% zirconium matrix composites reinforced with eggshell (ES). Using a milling machine, the constituent powders were combined, compressed at 550 MPa, and then sintered for two hours at 450 °C in an inert atmosphere. The structure, microstructure, densification, and microhardness of the produced composites were all evaluated. Hank's medium was used to study the performance of in vitro corrosion. With densifications of 99.65%, 98.9%, and 99.2%, the results revealed that the sintered samples of magnesium one-zirconium, magnesium two-zirconium, and magnesium one-zirconium-two had a relatively homogenous distribution of reinforcement. The findings showed that the sintered samples of magnesium-1% zirconium, magnesium-2.5% eggshell, and magnesium-1% zirconium-2.5% eggshell displayed a highly homogeneous reinforcement distribution with densifications of 99.65%, 98.9%, and 99.2%, respectively. The micro-hardness of the sintered magnesium-1% zirconium and magnesium-1% zirconium-2.5% eggshell samples rose by 13% and 6%, respectively, when compared to pure magnesium. The bio-corrosion experiment showed the benefit of reinforcing magnesium with zirconium and eggshell particles and their symbiotic impact by displaying greater polarization resistance and lower corrosion rate values in Hank's medium. The outcome demonstrates that eggshells can be used as effective, environmentally friendly reinforcing particles in the development of magnesium and magnesium–zirconium-based composites with improved in-phase corrosion properties for biomedical engineering.

## 3. Method for Developing Metal Matrix Composite

The stir casting method is a renowned method employed to produce metal matrix composites. It involves a homogeneous mechanical mix of the reinforcements, thereby causing a uniform distribution of the matrix and the reinforcements [81]. Kareem et al. [82] made it clear that the stir casting process is primarily employed to produce particulate reinforced metal matrix composite (PAMMC). It was also stated that "it is economical, simple, and the most widely used commercial technique known as the "vortex technique". According to Hashmi et al. [83], the stir casting method is a popular and cost-effective method for mass-producing metal matrix composites.

Saravanan et al. [84] suggested that the stir casting process is more effective in a two-step process involving heating the matrix above its liquidus temperature, allowing it to cool down between the solidus and liquidus temperatures, i.e., a semi-solid state, pouring your reinforcement particulate, then heating again. Sahu et al. [85] said that stir-casting technology is less expensive than other manufacturing technologies. He also stated that the method of stirring used, which is the stir casting technique, has a direct impact on the properties of composites. Inegbenebor et al. [86] stated that stir casting, a liquid processing sequence, is predominant due to its cost-effectiveness. At the same time, the processing parameters can be readily speckled and scrutinized. Figure 9 depicts the experimental setup of AMMC during the production process. Figure 10 illustrates the flow chart design for AMMC.

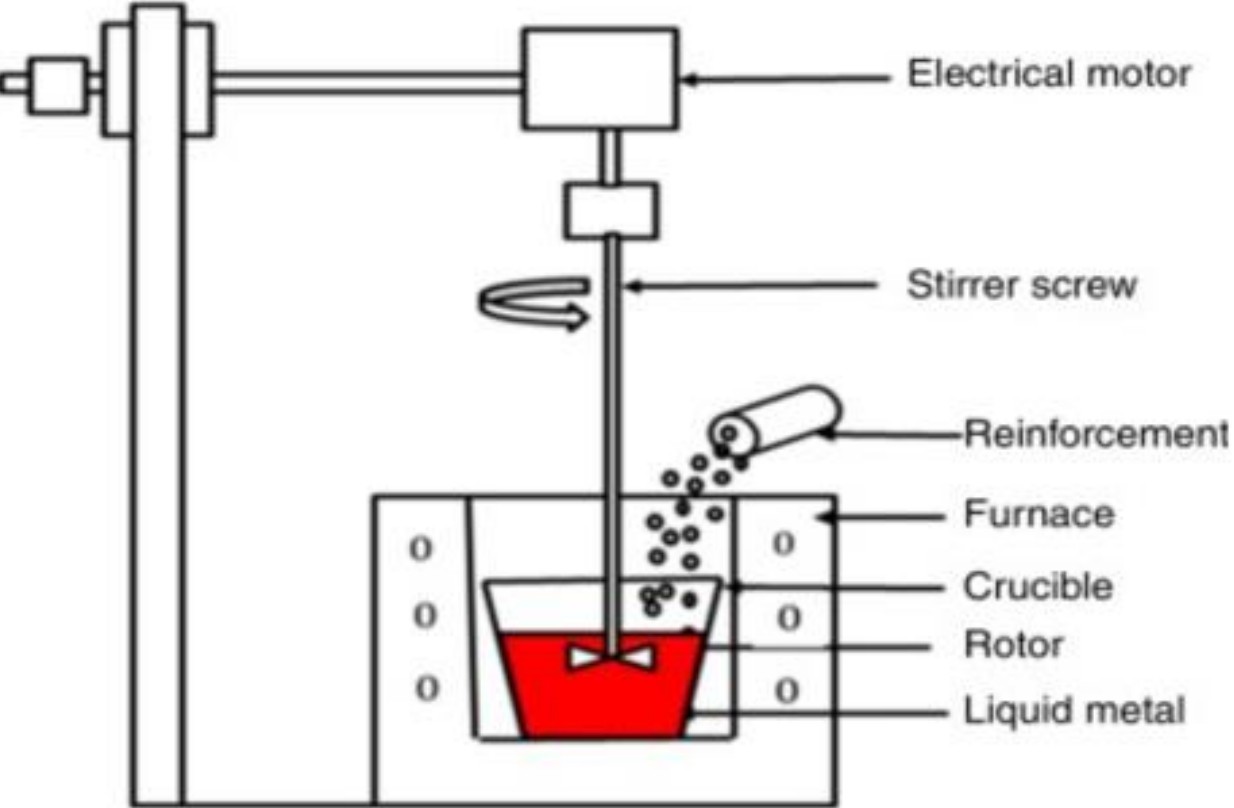

**Figure 9.** Stir casting system for AMMC manufacturing procedure.

Furthermore, Table 1 discusses the findings of other agro-waste materials that are viable for the reinforcement of aluminum alloy to produce a sustainable composite for the application of wind turbine blades.

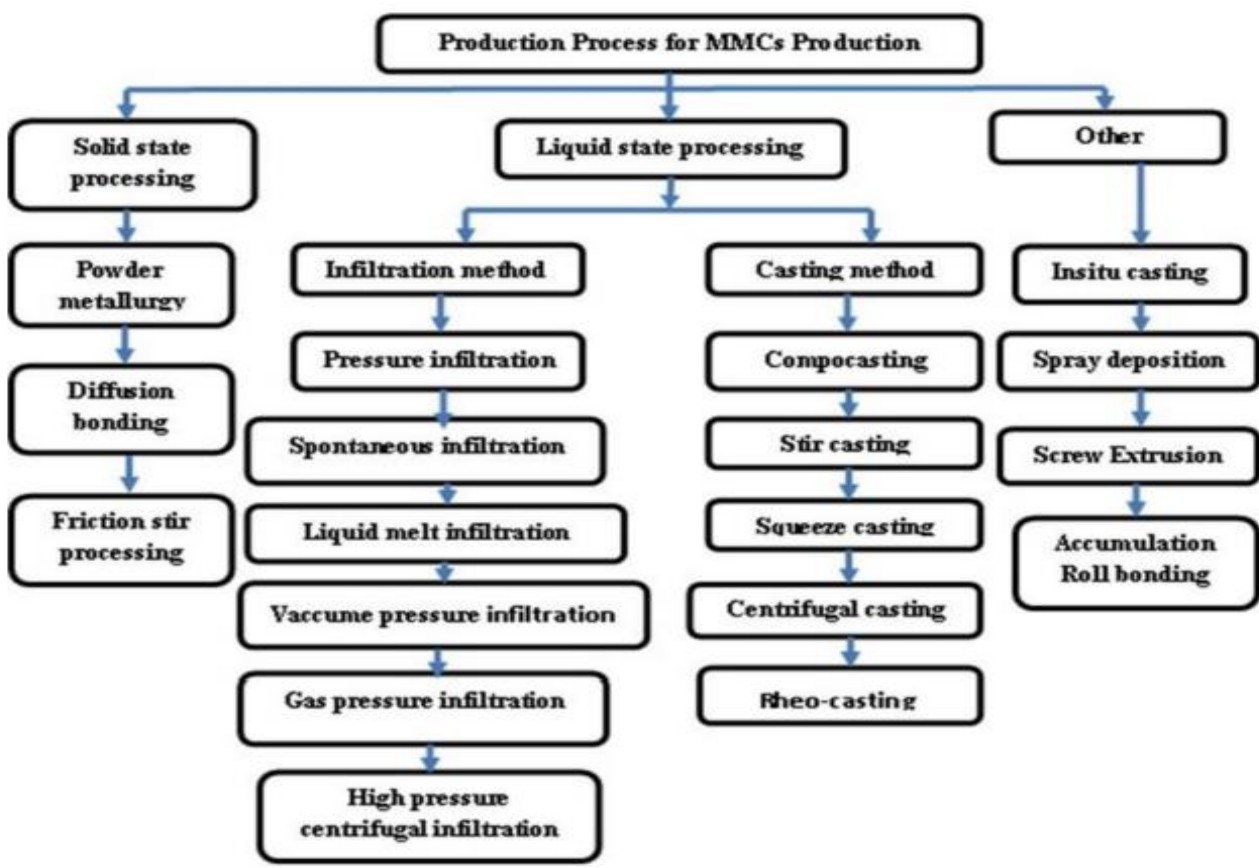

**Figure 10.** Process employed to produce AMMCs [87,88].

**Table 1.** Summary of a different method of fabrication of reinforcement, its composites developed, and the findings of the agro-base and eggshell on the hybrid aluminum metal matrix composite.

| Authors | Composite Matrix | Reinforcement and Coating | Method of Fabrication | Component Produced | Result(s) |
|---|---|---|---|---|---|
| Kaladgi et al. [89] | Aluminum 6061 | coconut shell particles as reinforcement and $Al_2O_3$ | stir casting technique | Al–alloy/$Al_2O_3$/coconut shell matrix composites | The findings show that, when compared to the base Al alloy, the stir-formed Al alloy $Al_2O_3$ and reinforced composites made of coconut shell ash are clearly superior in terms of compressive strength, hardness, impact strength, and torsional strength. |
| Tan et al. [90] | Aluminum 6061–SiC alloy | coconut shell particles | friction-stir processing | lightweight composite materials brake discs with an AA6082 alloy as the basis and a top layer made of A357/SiC AMMC | A significant increase in hardness and microstructure refinement is obtained in the regions immediately beneath the A357/SiC AMMC top layer, and elongation is maximized because the intermetallic particles have been refined by friction stir processing. |
| Kumar et al. [91] | AA7075 | $Al_2O_3$ and coconut shell ash (CSA) | two-stage stir casting with different reinforcement weight percentages (0–5) | AA7075 hybrid metal–matrix composites (HMMCs) | After the addition of $Al_2O_3$ and CSA-reinforced particles, it was shown that mechanical characteristics and tribological behavior had risen, but impact strength had somewhat decreased. |

| Authors | Composite Matrix | Reinforcement and Coating | Method of Fabrication | Component Produced | Result(s) |
|---|---|---|---|---|---|
| Panda et al. [92] | AA 1200 aluminum | coconut shell ash (CSA) | compo-casting route by melting in a stir-casting furnace | aluminium-coconut shell metal matrix Composite | The composites' wear rate declined till a modified CSA addition of 4 weight percent. The wear rate that rose as reinforcement was added further. The wear test findings and the hardness results were highly congruent; the highest hardness was discovered for Al–4wt% modified CSA. |
| Akbar et al. [93] | Al6061–Al$_2$O$_3$-SiC | agro-waste reinforcement such as rice husk ash, coconut shell ash (CSA), and sugar Bagasse ash | friction stir processing (FSP) | hybrid MMCs | Due to its usage as reinforcement for metal composites, studies on composite reinforcement from industrial and agricultural wastes demonstrate the waste's significant economic potential. Mechanical qualities are improved with the inclusion of reinforcement from industrial and agricultural wastes. |
| Harish et al. [94] | Aluminum alloy 5056 | Silicon Carbide of 3% and various amounts (2, 4%) of Bagasse ash from sugar cane | stir casting method | Aluminum alloy 5056 base matrix hybrids composites | The composites' wear rate and hardness value both significantly improved as a result of the findings. In addition to improving the qualities of the composite, the use of industrial agricultural wastes such as Bagasse ash as reinforcement encourages sustainability through waste management. |
| Purushothaman and Balakrishnan [95] | Al6061 alloy | coconut shell ash | stir casting method | Al6061–CSA composites | Comparing the resulting composites to the basis matrix aluminum alloy Al6061, corrosion resistance is improved. Up to 6 wt% of CSA content, the corrosion resistance of the composites begins to rise, and beyond that, it begins to fall. |
| Refaa et al. [96] | Al–Si–Mg matrix | peanut shell ash (PSA) | double stir casting technique | AA8079–PSA matrix composites | It was discovered that pockets of agglomerated reinforcement particles were scattered evenly throughout the aluminum matrix and were made of peanut shell ash reinforcements. PSA particles were added to the composite to improve its density, wear index, and hardness. |
| Mohan et al. [97] | Al6082 | eggshell and Si$_3$N | rriction stirs processing and post-process artificial ageing | Al6082-Si$_3$N$_4$—Mg-eggshell matrix composite | The findings indicated that the mechanical properties of reinforced materials may be improved. Additionally, it will be examined in light of the machining parameters used throughout the CNC turning process. When compared to normal aluminum, the analysis of variance from the optimization technique results in a notable increase in material removal rate (MRR) and a significant decrease in surface roughness (Ra) and machining time. |

Table 1 shows the summary of some materials, their composite reinforcements, their method of fabrication, the composite developed, and the effects of the agro-base and

eggshell on the developed matrix composite. A related study has also proven that reinforcement particulate is very essential in composite manufacturing [98–106]. Table 2 also presents related works in material development with other viable reinforcement materials. From Tables 1 and 2, it can be noted that the reinforcing materials are significant aspects of which it is necessary to prove the mechanical, chemical, corrosion resistance, electrical conductivity, and tribological properties of the materials needed to operate in high altitudes. Further, the hybrid production process increases the asperities of the materials.

**Table 2.** Summary of other reinforcement materials for metal matrix composite.

| Authors | Composite Matrix | Reinforcement and Coating | Method of Fabrication | Component Produced | Result(s) |
|---|---|---|---|---|---|
| Fayomi et al. [107] | AA8011 alloy | ZrB2–Si3N4 | two steps stir casting route | hybrid AA8011/ZrB2-Si3N4 nanomaterials | Additionally, the matrix alloy has a lower friction coefficient than composites and its friction coefficient decrease as ZrB2–Si3N4 percentage rises. When compared to AA8011, the wear rates of alloy/ZrB2-Si3N4 composites are lower, and they get even lower as the ZrB2–Si3N4 content rises. |
| Ghazanlou et al. [108] | Al7075 | graphene nanoplates/carbon nanotubes | stir casting process | Al7075/graphene nanoplates/carbon nanotubes composites | According to EBSD data, the presence of PSN at the reinforcements/matrix interface determines how much of the recrystallization occurs. Due to the substantial CTE mismatch between the GNPs and the matrix, GOS maps revealed the presence of PSN in all grains close to the GNPs. |
| Sankar et al. [109] | Al 6061 | Boron carbide–Graphite | N/A | N/A | The anodic dissolving process is hampered by higher graphite and boron carbide percentages. SiC abrasive media was added to the electrolyte flow route to alleviate the unfavorable effect on the anodic dissolution process. Performance in machining has improved thanks to sic particles. The experimental findings make it abundantly clear that the abrasive-assisted ECM produces performance that is generally superior to that of the ECM. |
| Moustafa et al. [110] | AA7075 aluminum alloy | hexagonal boron nitride, silicon carbide, tantalum carbide, and niobium carbide nanoparticles | friction stir process | N/A | Compressive strength and microhardness of the best hybrid alloy, AA7075/BN TaC, increased by 26.5% and 40%, respectively, compared to the basic alloy. |

**Table 2.** *Cont.*

| Authors | Composite Matrix | Reinforcement and Coating | Method of Fabrication | Component Produced | Result(s) |
|---|---|---|---|---|---|
| Prabhakar et al. [111] | LM14 aluminum alloy | B4C particles of 33μm size | stir casting process | aluminum/boron carbide metal matrix composite | Microstructural analysis showed that the matrix's particles were distributed uniformly, and tribological data demonstrated that the relationship between wear rate and coefficient of friction and load is direct, while the relationship between wear rate and coefficient of friction and sliding speed and distance is inverse. |
| Verma and Rao [112] | Al 6061 alloy | Boron carbide and rice husk ash | stir casting process | hybrid aluminum alloy 6061 based on rice husk ash and boron carbide | Comparing the addition of B4C to the addition of rice husk ash, a substantial improvement in hardness was seen (RHA). The hardest material was found at 5% B4C and 5% RHA. |

## 4. Al–Si–Mg Alloy Reinforced with Agro-Based and Eggshell

The study of the mechanical properties and the microstructural analysis of the hybrid reinforcement of metal matrix composites using agro-waste and eggshells will be reviewed in this section. Khan et al. [113] carried out a study on the microstructural and mechanical characteristics of a brand-new ternary-reinforced AA7075 hybrid metal matrix composite and compared them. The stir casting liquid metallurgical process was used to create four samples, including AA7075 (base alloy), AA7075–5wt%SiC (MMC), AA7075–5wt%SiC–3wt% rice husk ash (RHA), and AA7075–5wt%SiC–3wt%RHA-1wt% carbonized eggshell (CES). The experimental densities matched the theoretical values, demonstrating that the samples were successfully made. For the n-HMMC, a minimum density of 2714 kg/m$^3$ was noted. Additionally, n-HMMC had the highest porosity, which was found to be 3.11%. Additionally, the novel hybrid metal matrix composite (n-HMMC) demonstrated a 24.4% improvement in ultimate tensile strength and a 32.8% increase in hardness when compared to the basic alloy, as shown in Figure 11. The basic alloy AA7075 contains coarse grains with a minimum amount of porosity, as seen in Figure 12a. Figure 12b,c show some SiC agglomerates in the matrix material, whereas Figure 12d shows that n-HMMC contains a mixture of fine and coarse grains as well as some pores. It is important to note that poor mixing, pouring problems, and trapped gases are the most likely causes of porosity. The main component of CES is $CaCO_3$, which combines with oxygen to produce $CO_2$ gas. Because of the pores left behind, $CO_2$ escapes from the molten mixture and causes foaming. However, it has been found that the distribution of the particles in the matrix is improved when the 3 wt% RHA is added to the MMC to create s-HMMC. This result is in line with the study of Dixit and Suhane [114].

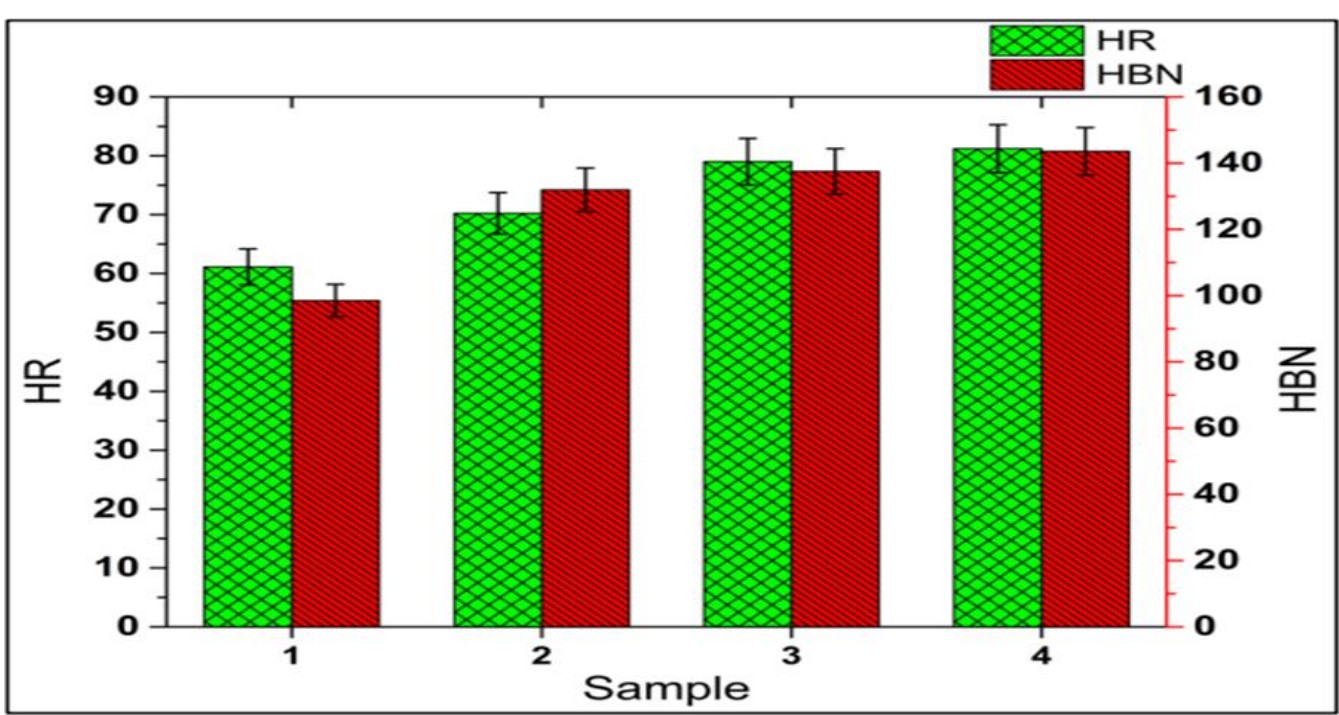

**Figure 11.** AA7075 (base alloy, sample 1), AA7075–5wt%SiC (MMC, sample 2), AA7075–3wt%RHA (s-HMMC, sample 3), and AA7075–5wt%SiC–3wt%RHA–1wt%CES all had average valves of ultimate tensile strength and percentage elongation (n-HMMC, sample 4).

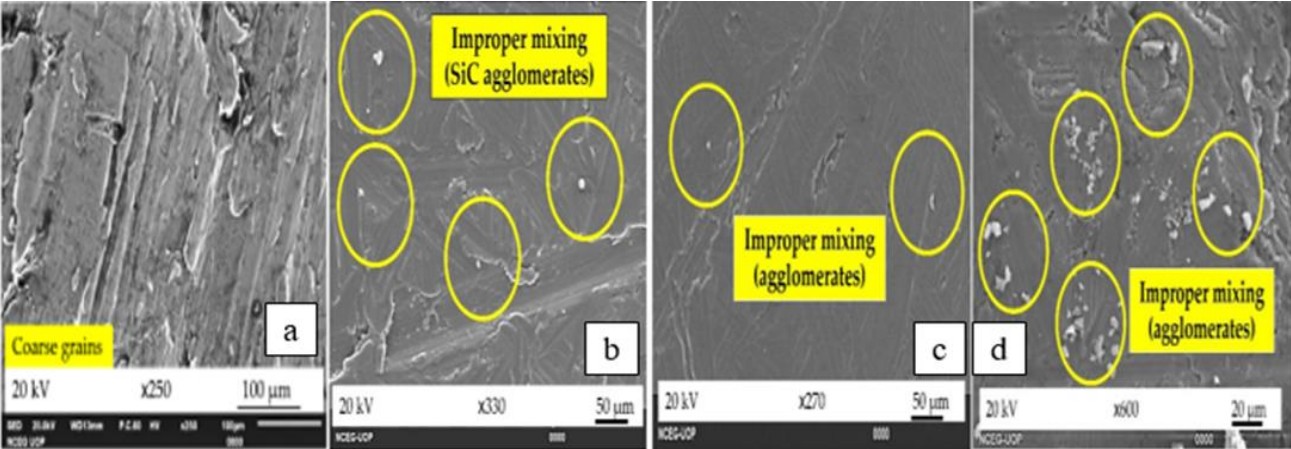

**Figure 12.** SEM image's microstructural study of: (**a**) AA7075 alloy, (**b**) AA7075–5wt%SiC, (**c**) AA7075–5wt%SiC–3wt%RHA, and (**d**) AA7075, 5 wt% SiC, 3 wt% RHA, 1 wt% CES.

Following the work of Olusesi and Udoye [115], this study demonstrates the possibilities of using agricultural waste materials as reinforcement rather than more conventional synthetic reinforcements such as silicon carbide and boron carbide. For this investigation, clay particles and rice husk ash are used because they have a track record of enhancing composite properties including hardness, tensile strength, and bend resistance. This study demonstrates the viability of agro-waste reinforcement in comparison to synthetic reinforcement through a variety of literature reviews and tests carried out at various reinforcement weight percentages. AA6061/clay + RHA composite was successfully manufactured using the stir casting method with various weight percent reinforcements of 2.5%, 5%, 7.5%, and 10%. Scanning electron microscope (SEM) (photographs demonstrate the composite's homogeneous dispersion of reinforced particles) and XRD (X-ray diffraction analysis) images

illustrate the composite's occurrence of distinct phases. In terms of hardness properties, the 7.5 wt% reinforcement was superior to other composite weight reinforcements, as shown in Figure 13. According to the results of the SEM/EDS study, the matrix showed a uniform distribution of reinforcement.

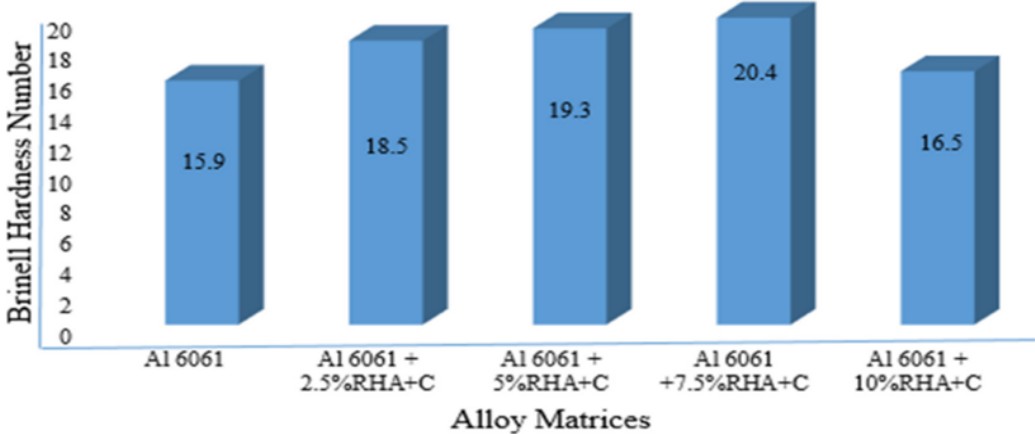

**Figure 13.** The impact of clay concentration and RHA on microhardness of the alloy matrix.

Gupta et al. [116] examine the wear behaviour and dry sliding friction of AA7075 composites reinforced with carbonized eggshells and rice husk ash. The stir casting method was used to create composites with varied weight percentages of carbonized eggshells and rice husk ash in the range of 0 to 5 wt.%. Prior to tribological testing, density, porosity content, and microhardness were computed. On a pin-on-disc type tribometer operating at room temperature, tests for friction and wear were performed. The samples were put to the test under various loads (10–50 N), at a constant speed of 1 m/s, and at a constant sliding distance of 1500 m. Additionally, samples were evaluated with a constant normal load of 30 N at two different sliding speeds (3 and 5 m/s). The density of composites was lowered by the inclusion of natural reinforcements. The sample's microhardness rose by 15.08% above the base composition when rice husk ash (5 wt%) was added. A sample with 5 wt% rice husk ash demonstrated the highest wear resistance. The sample with 3.75 wt% rice husk ash and 1.25 wt% carbonized eggshells displayed the highest coefficient of friction. While the coefficient of friction increased with increasing load and decreased with increasing sliding speed for all composites, wear loss varied directly with increasing load and sliding speeds. Scanning electron microscopic pictures of worn surfaces show that ploughing and delamination are the primary wear mechanisms at low speed, while ploughing is the dominant wear mechanism at high speed. For many tribological applications, particularly in the automotive industry, composites with increased wear-resistant qualities can be used.

According to Idusuyi et al. [117], one of the difficulties with employing aluminum-based MMCs is understanding the impact of the reinforcing particles on the corrosion resistance and mechanical qualities. In this study, the mechanical characteristics and corrosion behaviur of Al6063 reinforced with eggshell ash and rice husk ash were examined. The stir casting technique was used to make composite materials with 10% reinforcements from waste eggshell ash (ESA) and rice husk ash (RHA) that were each 212 m in size. In the following ratios—10:0, 7.5:2.5, 5:5, 2.5:7.5, and 0:10—the RHA and ESA were added. Baseline material was unreinforced Al6063. The composites were characterized using immersion tests, potentio-dynamic polarization methods, tensile testing, optical microscopy (OM), and scanning electron microscopy (SEM). In general, a decrease in corrosion rates was seen for reinforced composites as the weight percentage of RHA rose. As the amount of ESA in the matrix increased, the composites' porosity levels decreased [118]. Islam et al. [119] carried out a study on eggshells and rice husks as sustainable reinforcement materials for an aluminum alloy to study the produced composite material's corrosion behaviour, grain size, and coefficient of thermal expansion (CTE). The results show that the corrosion rate of

the composite, tensile strength, and hardness are enhanced due to the dispersion of eggshell particles in phases in the metal matrix [120,121]. According to Okokpujie and Tartibu [122], corrosion is a crucial factor that is considered during the production process. There is a need to ensure that the materials can withstand corrosion, as the development of new metal composites has shown to perform remarkably well in terms of weight-to-strength ratio and mechanical characteristics improvement. Yet, a lot of material reinforcing and corrosion inhibitors need to be researched and used to combat corrosion. This investigation will concentrate on how aluminum metal matrix composites (AMMC) reinforced with coconut rice and eggshell will behave when exposed to 0.4 M $H_2SO_4$. The created aluminum metal matrix composite was protected by $TiO_2$ nanoparticles in the study from the acidic environment of $H_2SO_4$ by acting as an inhibitor. In this study, weight loss analysis was also done to study the corrosion rate and the inhibitor efficiency, and SEM and EDX were used to study the surface morphology of the corroded samples in acidic environments. Samples C and E are made of Al 8112 alloy, 2.5% coconut rice, and 2.5% eggshell, and 5% coconut rice and 3% eggshell, respectively. The result shows that sample E with 5% coconut rice and 3% eggshell has the lowest corrosion rate compared to samples D and C due to the impact of the coconut rice and eggshell which have a high percentage of carbon and silicon element. The study of eggshell and coconut rice reinforcement carried out by Okokpujie et al. [123] shows that the reinforcement significantly improves the mechanical, electrical, and corrosion properties of the aluminum composite.

## 5. Conclusions

This study reviewed the sustainability of the implementation of the composition of aluminum, silicon, and magnesium with agro-waste base (coconut rice, coconut shell, rice husk asks, Bagasse ash from sugar cane), and eggshell for the development of matrix composite as an ecological material for wind turbine blade production. In addition, the study reviewed composite preparation techniques and provided a summary of the applications of other existing reinforcement materials in the development of wind turbine blade materials. There are significant findings from several kinds of literature on the matrix composite reinforced with agro-waste, silicon carbide, magnesium, and the strong binding force of the eggshell as follows:

i.    Matrix composites prepared with aluminum alloy as the base material are viable for producing high-quality materials for wind turbine blade applications. Aluminum alloys are widely used because they have high strength and low weight.

ii.   The combination of magnesium and silicon with an aluminum alloy increases the mechanical properties, such as the high resistance to wear rate during operations and increases the hardness and ductility of the material.

iii.  The matrix composition done with the implementation of the eggshell has been found to possess high resistance to corrosion, which is highly needed in the materials to develop wind blades. Because of the high speed at which wind blades operate in moist conditions,

iv.   The application of agro-based reinforcement such as coconut shell, coconut rice, rice husk, and bamboo roots, as shown in this review, shows that it is viable and cost-effective to produce sustainable materials for engineering applications.

## 6. Recommendation

After carrying out an extensive review on the application of aluminum, silicon, magnesium, agro-base, and eggshell to develop a matrix composite for a sustainable material for wind blade manufacturing, this study will recommend the following due to the critical condition of the application of the wind blade:

i.    A viable study should be conducted to produce a quaternary composite that can work in multi-faceted ways since the wind turbine blade is sectioned with different structural areas;

ii.  The concentration percentage of the various combinations of the study materials should be studied to come up with the optimal ratio that can work very well by building a prototype to test the functionality of the materials developed.

**Author Contributions:** Conceptualization, I.P.O. and L.K.T.; methodology, I.P.O. and L.K.T. software, I.P.O. and L.K.T.; validation, I.P.O. and L.K.T.; formal analysis, I.P.O. and L.K.T.; investigation, I.P.O. and L.K.T.; resources, I.P.O. and L.K.T.; data curation, I.P.O. and L.K.T.; writing—original draft preparation, I.P.O. and L.K.T.; writing—review and editing, I.P.O. and L.K.T.; visualization, I.P.O. and L.K.T.; supervision, I.P.O. and L.K.T.; project administration, I.P.O. and L.K.T.; funding acquisition, I.P.O. and L.K.T. All authors have read and agreed to the published version of the manuscript.

**Funding:** This research received no external funding.

**Data Availability Statement:** This study did not have any data.

**Conflicts of Interest:** The authors declare no conflict of interest.

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
