# Peer review of "Aluminum Alloy Reinforced with Agro-Waste, and Eggshell as Viable Material for Wind Turbine Blade to Annex Potential Wind Energy: A Review"

_jcs, doi:10.3390/jcs7040161_

Round 1

Reviewer 1 Report (Previous Reviewer 3)

The resubmitted manuscript has been well organized. The readers can get useful information from this review. 

Author Response

The resubmitted manuscript has been well organized. The readers can get useful information from this review. 

Thanks so much, reviewer 1. we appreciate your comment.  

Reviewer 2 Report (Previous Reviewer 2)

paper is well written and it gives very good research gap for the new research areas.

Grammatical mistakes and word discontinuity observed in some of the places.

Paper should be throughly checked before publication for grammatical errors.

Author Response

RESPONSE TO REVIEWER’S COMMENTS

1Q paper is well written and it gives a very good research gap for the new research areas.

Response Thank you reviewer 2

2Q Grammatical mistakes and word discontinuity were observed in some of the places.

Response, the authors have worked on the grammatical error. We believe this version is suitable for publication.

The paper should be thoroughly checked before publication for grammatical errors.

Response: The check has been done

Thanks

Round 2

Reviewer 2 Report (Previous Reviewer 2)

Paper can be accepted in the present format

This manuscript is a resubmission of an earlier submission. The following is a list of the peer review reports and author responses from that submission.

Round 1

Reviewer 1 Report

Dear Authors

The topic of the study shows that the use of a composition of Al-Si-Mg alloy with coconut rice, coconut shells and eggshells improves the strength of the material. Very little is devoted to this thesis in the study.

Literature analysis should refer to the paper topic. While in this case it include general characteristics of the components of composite materials. The summary is a list of manufacturing methods and types of composites not necessarily used in this type of equipment.

The conclusions are vague and imprecise.

Kind regards

Author Response

RESPONSE TO REVIEWER 1 COMMENTS

The authors use this medium to appreciate the reviewers and chief editors for the comprehensive review of this manuscript. Below is the response to the reviewer one (1) comments.

Question 1: The topic of the study shows that the use of a composition of Al-Si-Mg alloy with coconut rice, coconut shells and eggshells improves the strength of the material. Very little is devoted to this thesis in the study.

Response 1: Thank you for the reviewer 1. The authors have worked on the manuscript and has improve it with relevant paper as pertain to the topic

Question 2: Literature analysis should refer to the paper topic. While in this case it includes general characteristics of the components of composite materials. The summary is a list of manufacturing methods and types of composites not necessarily used in this type of equipment.

Response 2: This section explains the method of fabrication, the composite developed, and findings of the agro base and egg shell on the developed composite.

Question 3: The conclusions are vague and imprecise.

Response 3: with extensive work on the manuscript, the conclusion is very okay and not vague

Thanks

Reviewer 2 Report

The paper is well written by considering all the past work and it is very useful for all the researchers.

Novality of the work also well explained.

Author Response

RESPONSE TO REVIEWER 2 COMMENTS

The authors use this medium to appreciate the reviewer 2 and chief editors for the comprehensive review of this manuscript. 

Question 1: The paper is well written by considering all the past work, and it is very useful for all the researchers.

Response 1: Thank you, reviewer 2, for the review and encouragement towards the improvement of this manuscript.

Question 2: Novelty of the work also well explained.

Response 2: Thanks

Reviewer 3 Report

The paper titled “Al-Si-Mg Alloy Reinforced with Agro-Based, and Eggshell for Development of Matrix Composite for Wind Turbine Blade for Viable Wind Energy Production: A Review” concluded some reinforced elements or particles from the natural materials. It is significant due to the sustainability and economy when used the low-cost materials. However, there are still some contents worthy of discussion in the paper.

1. From the title and the abstract, the readers think this review may discuss the Al-Si-Mg Alloy reinforced by different elements from crops or eggshell. However, the part-title 2.2 Magnesium as a sustainable reinforcement material viewed the magnesium-based MMCs, which was not consistent with the topic. Absolutely, the paper should discuss the effects on Al-Mg-Si alloy resulting from other elements or somethings based on the coconut rice, coconut shell and egg shell except Mg and Si, because the Mg and Si have been the major elements, and the effects on properties of Aluminum has been fully reported.  

2. P273-297 reviewed the magnesium-based MMCs, however, this part titled the “2.1 Aluminum Alloys as a sustainable reinforcement material”. It is not proper to mention the magnesium-based MMCs in 2.1.

3. In 2.1, there was too many descriptions for the production of aluminum. The readers need more information about the Aluminum-based MMCs.

4. Figs. 6 and 7 gives the coconut and eggshell. It is not a useful expression because the readers cannot get more information only from the simple pictures that everybody all know. The readers need the summarizations about how the coconut or eggshell affects the properties of Aluminum-based MMCs.

  5. Figs. 9 and 10 gives the common methods for preparing MMCs. There was lack of the specific preparation process and the technical difficulty for Aluminum-based MMCs, and also lack of the comparisons between these methods.

  6. As mentioned comment 1, although the Table 1 summarized the different method of fabrication of reinforcement materials for hybrid aluminum metal matrix composite, the authors represented quite a small data of Al-Mg-Si based MMCs.

  In a word, the paper title makes people not understand what the authors want to summarize.

Author Response

RESPONSE TO REVIEWER 3 COMMENTS

  1. From the title and the abstract, the readers think this review may discuss the Al-Si-Mg Alloy reinforced by different elements from crops or eggshell. However, the part-title 2 Magnesium as a sustainable reinforcement material” viewed the magnesium-based MMCs, which was not consistent with the topic. Absolutely, the paper should discuss the effects on Al-Mg-Si alloy resulting from other elements or somethings based on the coconut rice, coconut shell and egg shell except Mg and Si, because the Mg and Si have been the major elements, and the effects on properties of Aluminum has been fully reported.  

Response 1: Thank you, reviewer 3, however. The authors feel that when discussing and hybrid materials, it is necessary to also review works on the individual materials like Silicon, Magnesium. So it is okay discussing it. Thanks

  1. P273-297 reviewed the magnesium-based MMCs, however, this part titled the “1 Aluminum Alloys as a sustainable reinforcement material”. It is not proper to mention the magnesium-based MMCs in 2.1.

Response 2: Aluminum alloys can be reinforced and can also be used as reinforcement in several area of research. So from the authors view, these subsections are viable.

  1. In 2.1, there was too many descriptions for the production of aluminum. The readers need more information about the Aluminum-based MMCs.

Response 3: This is okay by the view of the authors. However, the authors has worked on the manuscript to improve it. Thanks

  1. Figures 6 and 7 gives the coconut and eggshell. It is not a useful expression because the readers cannot get more information only from the simple pictures that everybody all know. The readers need the summarizations about how the coconut or eggshell affects the properties of Aluminum-based MMCs.

Response 4: Please note that coconut rice is a newly introduced reinforcement materials introduced by the authors in this research work because the authors have an ongoing research work presently using it. The SEM AND EDS is done by the authors. So there is no much work on coconut rice reported. What is recorded is the output from our lab. Results.

  1. Figures 9 and 10 gives the common methods for preparing MMCs. There was lack of the specific preparation process and the technical difficulty for Aluminum-based MMCs, and also lack of the comparisons between these methods.

Response 5: However, Table 1 and Table 2 show different method for the MMCs development methods.

  1. As mentioned comment 1, although the Table 1 summarized the different method of fabrication of reinforcement materials for hybrid aluminum metal matrix composite, the authors represented quite a small data of Al-Mg-Si based MMCs.

Response 6: A section has been created to discussed this, and the author has worked on the Al-Mg-Si based MMCs section 4 in the manuscript.

  1. In a word, the paper title makes people not understand what the authors want to summarize.

Response 7: Thank you, reviewer 3. The work and the title of the manuscript is well correlated. 

Thank you for the review.

Round 2

Reviewer 1 Report

Dear Authors,

The corrections made are not sufficient to accept the paper. The title of the paper mentions the reinforcement of the Al-Si-Mg Agro-Based, and Eggshell alloy, and the Authors cite tests on the Al/SiC + Agro-Based, and Eggshell composite.

The article was supplemented with test results: Al/SiC composite + Agro-bassed and Eggshell; Figure 11 (should be marked 12) shows a micrograph of composites where it is difficult to identify the reinforcing phase, only agglomerates are indicated.

What size are the SiC reinforcing particles in suspension?

What size were the Agro-bassed and Eggshell particles introduced into the Al/SiC suspension?

What is SiC on the micrographs and what is Agro-bassed and Eggshell?

Why, after carbonized the Eggshell (fig. 8), the content of elements: Fe - 11% and Si - 63%? The Eggshell consists of approx. 98.5% of calcium carbonate, the rest is trace elements. It is well known that Fe has a negative effect on aluminum alloys.

Kind regards

Reviewer 3 Report

Actually, it is interesting to investigate the effects of the new reinforcements, such as agro-based materials on the aluminum matrix composite. However, for this paper itself, the authors should review the researches about the effects of these new reinforcements on the properties or microstructures on Al-Si-Mg alloy. Please note that, the Al-Si-Mg alloy is a ternary alloy and the matrix. Why the authors mentioned so many to describe the Al-based,Mg-based and Si-based alloy alone?

Although the v2 version add “4. Al-Si-Mg Alloy Reinforced with Agro-Based and Eggshell”, it has not fundamentally change of the organizational structure of the whole paper.

In addition, there are two Fig.11 in page 23, and the 7075 aluminum alloy are also not a Al-Si-Mg alloy.